A new basal ichthyosauromorph from the Lower Triassic (Olenekian) of Zhebao, Guangxi Autonomous Region, South China

Ren Jicheng 1
http://orcid.org/0000-0001-9636-0307 Jiang Haishui 2
Xiang Kunpeng 3
http://orcid.org/0000-0002-5488-6797 Sullivan Corwin 4 5
He Yongzhong 3
Cheng Long 6
Han Fenglu 2 hanfl@cug.edu.cn
1 School of Li Siguang, China University of Geosciences (Wuhan) , Wuhan, Hubei Province , China
2 School of Earth Sciences, China University of Geosciences (Wuhan) , Wuhan, Hubei Province , China
3 Guizhou Geological Survey , Guiyang, Guizhou Province , China
4 Department of Biological Sciences, University of Alberta , Edmonton , Canada
5 Philip J. Currie Dinosaur Museum , Wembley , Canada
6 Wuhan Centre of China Geological Survey , Wuhan , China
Hedrick Brandon
Electronic publication date: 2022 Apr 7
Publication date: 2022
Volume: 10
Electronic Location ID: e13209
Received 2021 May 20; Accepted 2022 Mar 10
Copyright: © 2022 Ren et al.
Copyright year: 2022
Copyright holder: Ren et al.
License: This is an open access article distributed under the terms of the Creative Commons Attribution License, which permits unrestricted use, distribution, reproduction and adaptation in any medium and for any purpose provided that it is properly attributed. For attribution, the original author(s), title, publication source (PeerJ) and either DOI or URL of the article must be cited.
License URL: https://creativecommons.org/licenses/by/4.0/

Keywords: Early Triassic, China, Morphology, Ichthyosauromorpha

Funding: National Natural Science Foundation of China 41830320; 41688103; 41972014 Guizhou Scientific and Technology Planning Project QKHPTRC [2018]5626 Guizhou Bureau of Geology and Mineral Exploration and Development QDKKH [2020]29 Natural Sciences and Engineering Research Council of Canada RGPIN-2017-06246 This project was supported by the National Natural Science Foundation of China (grant no. 41830320; 41688103; 41972014), the Guizhou Scientific and Technology Planning Project (QKHPTRC [2018]5626), the Scientific Research Project of the Guizhou Bureau of Geology and Mineral Exploration and Development (QDKKH [2020]29), and the Natural Sciences and Engineering Research Council of Canada (Discovery Grant RGPIN-2017-06246). The funders had no role in study design, data collection and analysis, decision to publish, or preparation of the manuscript.

==============================
Here we describe a newly discovered basal ichthyosauromorph from the Lower Triassic of South China, Baisesaurus robustus gen. et sp. nov. The only known specimen of this new species was collected from the Lower Triassic (Olenekian) Luolou Formation in the Zhebao region of Baise City, on the northwest margin of the Nanpanjiang Basin, and comprises a partial skeleton including the ribs, the gastralia, a limb element, 12 centra, and seven neural arches. Comparisons to a wide variety of Early Triassic marine reptiles show Baisesaurus robustus to be a basal ichthyosauromorph based on the following features: neural arches lack transverse processes; dorsal ribs are slender, and not pachyostotic even proximally; and median gastral elements have long, sharp anterior processes. The limb element is long and robust, and is most likely to be a radius. Baisesaurus robustus is large (estimated length more than 3 m) relative to early ichthyosauromorphs previously discovered in China, and shares noteworthy morphological similarities with Utatsusaurus hataii, particularly with regard to body size and the morphology of the probable radius. Baisesaurus robustus also represents the first record of an Early Triassic ichthyosauromorph from Guangxi Autonomous Region, extending the known geographic distribution of ichthyosauromorphs in South China.

Introduction

Mesozoic marine reptiles first appeared in the Early Triassic, primarily in the form of three main clades: the sauropterygians, the thalattosaurs and the ichthyosauromorphs (Scheyer et al., 2014). Early Triassic sauropterygians include Corosaurus from North America (Storrs, 1991; Rieppel, 1998a), Cymatosaurus from Europe (Rieppel, 2000), and multiple genera from southern China, namely Kwangsisaurus (Young, 1959; Rieppel, 1998b) from Guangxi Autonomous Region (or Guangxi), Hanosaurus (Young & Dong, 1972a; Rieppel, 1998c) and Lariosaurus (Li & Liu, 2020) from Hubei Province, and Majiashanosaurus (Jiang et al., 2014) from Anhui Province. These taxa were coast-dwelling, slender-bodied animals that would have superficially resembled swimming lizards when they were in the water, and were not greatly specialized for marine life. Thalattosaurs are a less diverse group of small to medium-sized secondary marine reptiles (Scheyer et al., 2014). They were modestly adapted to an aquatic lifestyle, and have characteristic anatomical features including a long snout and a small, or even absent, upper temporal fenestra (Müller, 2005). Thalattosaurs are mainly known from the Middle and Upper Triassic of Europe (Müller, Renesto & Evans, 2005), North America (Nicholls, 1999) and China (Jiang et al., 2004). A few specimens of Thalattosaurus, Paralonectes and Agkistrognathus have been reported from the Lower to Middle Triassic Sulphur Mountain Formation in Canada, but their exact age is still uncertain (Nicholls & Brinkman, 1993b; Scheyer et al., 2014).

Ichthyosauromorpha (Motani et al., 2015a) is an expansive group that includes the two clades Hupehsuchia (Young & Dong, 1972b) and Ichthyosauriformes. Hupehsuchia is a distinctive clade of Early Triassic marine reptiles so far known only from five southern Chinese species, namely Nanchangosaurus suni (Wang, 1959), Hupehsuchus nanchangensis (Carroll & Dong, 1991), Parahupehsuchus longus (Chen et al., 2014b), Eohupehsuchus brevicollis (Chen et al., 2014c) and Eretmorhipis carrolldongi (Chen et al., 2015). Ichthyosauriformes includes the Nasorostra, represented by two short-snouted genera (Motani et al., 2015a; Jiang et al., 2016), along with four species of Chaohusaurus and the more diverse Ichthyopterygia. Most known ichthyopterygians belong to the derived clade Ichthyosauria, a group of Mesozoic marine reptiles with large eyes, fish-shaped bodies, and numerous other adaptations to the aquatic environment (Sander, 2000; Motani, 2009). Ichthyosauromorphs first appeared in the Spathian subage of the Early Triassic (Motani et al., 2015a; Moon, 2019), and maintained a cosmopolitan distribution until their extinction in the Cenomanian Age of the Late Cretaceous (Zammit, 2012). The origin and early evolution of ichthyosauromorphs are still under study, but many recent discoveries have greatly expanded scientific knowledge of Early Triassic members of the group (Motani et al., 2015a; Jiang et al., 2016; Huang et al., 2019; Moon, 2019). In this paper, non-ichthyosaurian ichthyosauromorphs are referred to as basal ichthyosauromorphs, and non-ichthyosaurian ichthyosauriforms are referred to as basal ichthyosauriforms.

In addition to the five hupehsuchian genera mentioned above, a variety of other basal ichthyosauromorph genera have been also reported from the Lower Triassic (Table 1). Most of these ichthyosauromorphs are from South China (Young & Dong, 1972c; Chen et al., 2013; Motani et al., 2015a; Jiang et al., 2016; Huang et al., 2019), Spitsbergen (Wiman, 1910; Wiman, 1929; Callaway & Massare, 1989; Maisch & Matzke, 2003a; Maisch, 2010) or Canada (Nicholls & Brinkman, 1995; Cuthbertson, Russell & Anderson, 2013), although others are from Thailand (Mazin et al., 1991) or Japan (Shikama, Kamei & Murata, 1978).

Table 1 Some representative ichthyosauromorphs from the Early Triassic.

Locality	Taxon	Stratigraphic horizon	References	Remarks	
Spitsbergen, Norway	Grippia longirostris	Vikinghøgda Formation	Wiman (1929), Motani (1997), Motani (2000), Hansen, Hammer & Nakrem (2018)	–	
Pessopteryx nisseri	Vikinghøgda Formation	Wiman (1910), Maisch (2010), Engelschiøn et al. (2018)	=Merriamosaurus hulkei
(Maisch & Matzke, 2002; Maisch & Matzke, 2003a; Maisch, 2010)	
Quasianosteosaurus vikinghoegdai	Vikinghøgda Formation	Maisch & Matzke (2003b)	–	
Omphalosaurus merriami	Vikinghøgda Formation	Wiman (1910), Maisch (2010), Sander & Faber (2003), Ekeheien et al. (2018)		
Isfjordosaurus minor	Vikinghøgda Formation	Wiman (1910), Motani (1999)	Only a humerus	
British Columbia, Canada	Parvinatator wapitiensis	Sulphur Mountain Formation	Nicholls & Brinkman (1995)	–	
Gulosaurus helmi	Sulphur Mountain Formation	Cuthbertson, Russell & Anderson (2013)	–	
Grippia sp.	Sulphur Mountain Formation	Brinkman, Zhao & Nicholls (1992)	–	
Utatsusaurus sp.	Sulphur Mountain Formation	Nicholls & Brinkman (1993a)	–	
Anhui, China	Chaohusaurus geishanensis	Nanlinghu Formation	Young & Dong (1972c)	–	
Chaohusaurus chaoxianensis	Nanlinghu Formation	Chen (1985), Motani et al. (2015b)	=Chensaurus chaoxianensis
(Mazin et al., 1991)
=Anhuisaurus chaoxianensis
(Chen, 1985)	
Chaohusaurus brevifemoralis	Nanlinghu Formation	Huang et al. (2019)	–	
Cartorhynchus lenticarpus	Nanlinghu Formation	Motani et al. (2015a)	–	
Sclerocormus breviceps	Nanlinghu Formation	Jiang et al. (2016)	–	
Hubei, China	Chaohusaurus zhangjiawanensis	Jialingjiang Formation	Chen et al. (2013)	–	
Miyagi, Japan	Utatsusaurus hataii	Osawa Formation	Shikama, Kamei & Murata (1978)	–	
Southern Peninsula,
Thailand	Thaisaurus chonglakmanii	Unrecorded horizon	Mazin et al. (1991), Liu et al. (2018)	Well preserved but not yet described in detail	

The upper Lower Triassic Vikinghøgda Formation in Spitsbergen has yielded many ichthyosauromorph specimens (Table 1). The skull and forelimbs of the small (1−1.5 m in length) species Grippia longirostris have been well studied (Wiman, 1929; Motani, 1997, 1998, 2000; Hansen, Hammer & Nakrem, 2018). Omphalosaurus, considered to be a durophagous animal, is represented by Vikinghøgda material that is indeterminate at the species level (Ekeheien et al., 2018). Quasianosteosaurus vikinghoegdai is an ichthyosauriform known from an unusual, relatively complete skull (Maisch & Matzke, 2003b), whereas Isfjordosaurus minor (Wiman, 1910; Motani, 1999) is represented only by an isolated humerus (Maisch, 2010). Pessopteryx nisseri is relatively large (Wiman, 1910), and is diagnosed by traits of the humerus (Motani, 1999; Maisch, 2010; Engelschiøn et al., 2018). Material referable to Cymbospondylus sp. has also been recovered from the Lower Triassic of Spitsbergen (Engelschiøn et al., 2018). Both Cymbospondylus and Pessopteryx are considered to belong within the derived clade Ichthyosauria (Moon, 2019).

In Canada, ichthyosauromorphs are found in the Lower to possibly Middle Triassic Sulphur Mountain Formation in the Wapiti Lake area, British Columbia (Scheyer et al., 2014). The Sulphur Mountain ichthyosauromorphs include Parvinatator wapitiensis (Nicholls & Brinkman, 1995), Gulosaurus helmi (Cuthbertson, Russell & Anderson, 2013), and specifically indeterminate specimens of Grippia and Utatsusaurus (Brinkman, Zhao & Nicholls, 1992; Nicholls & Brinkman, 1993a).

Early Triassic ichthyosauromorphs are abundant in South China. Setting aside the five hupehsuchian taxa, they mainly represent four species belonging to the genus Chaohusaurus. Among them, Chaohusaurus geishanensis (Young & Dong, 1972c), C. chaoxianensis (Chen, 1985; Mazin & Sander, 1993; Motani & You, 1998; Motani et al., 2015b) and C. brevifemoralis (Huang et al., 2019) are from the Nanlinghu Formation of Anhui Province, whereas the large fourth species, C. zhangjiawanensis, is known from well-preserved material from the Jialingjiang Formation of Hubei Province (Chen et al., 2013). Two additional Early Triassic ichthyosauriforms from South China, Sclerocormus breviceps (Jiang et al., 2016) and Cartorhynchus lenticarpus (Motani et al., 2015a), both have a short skull and a robust forelimb. They have been recovered in phylogenetic analyses as sister taxa forming the basalmost ichthyosauriform clade Nasorostra (Jiang et al., 2016; Huang et al., 2019). In addition, recent research has identified some similarities in dental morphology between Cartorhynchus and Omphalosaurus (Huang et al., 2020), which may imply a close relationship between Nasorostra and Omphalosauridae (Qiao et al., 2021).

The ichthyopterygian Utatsusaurus hataii from Japan (Shikama, Kamei & Murata, 1978) differs from Chaohusaurus in being a larger animal with a more robust forefin skeleton. In a recent phylogenetic analyses (Jiang et al., 2016; Moon, 2019), Utatsusaurus was found to be among the basalmost ichthyopterygians. Thaisaurus chonglakmanii (Mazin et al., 1991) might be even more primitive within Ichthyopterygia than Utatsusaurus, but is too incompletely known for its phylogenetic position to be certain (Liu et al., 2018).

Early Triassic ichthyosauriforms can generally be divided into two morphological categories (Zou et al., 2020). Type A includes Chaohusaurus geishanensis, C. zhangjiawanensis, Utatsusaurus hataii, Parvinatator wapitiensis, Grippia longirostris and Gulosaurus helmi. These taxa are relatively large (some exceeding 1 m in length), have elongate zeugopodial elements and a comparatively compact forelimb bone arrangement, and are considered to have been strong swimmers capable of long-distance migration, allowing them to disperse and diversify on a worldwide scale. Type B includes C. chaoxianensis (and possibly also C. brevifemoralis), Cartorhynchus lenticarpus and Sclerocormus breviceps, smaller forms (body length less than 70 cm) that share a short, anteroposteriorly broad, and relatively flexible forelimb. They are considered to have been weaker swimmers that had little capacity for migration, explaining why taxa of this type occur only in Anhui Province, China (Zou et al., 2020).

In 2018, a Guizhou Geological Survey field crew collected a partial skeleton of a new marine reptile, which is recognizable as an ichthyosauromorph, from the Lower Triassic Luolou Formation of the northwest margin of the Nanpanjiang Basin in Zhebao Township, Longlin County, Baise City, Guangxi, China (Fig. 1).

Figure 1 Location of fossil site.

The specimen was found in Zhebao Township, Longlin County, northwest part of Baise City, Guangxi Autonomous Region.

The Luolou Formation in the Nanpanjiang Basin consists of limestone rich in ammonoids and conodonts, deposited on a shallow marine carbonate shelf. In the Zhebao section, the formation is 98 meters thick and can be divided into 22 beds from bottom to top. The specimen described here was found in bed 14, which comprises a light gray medium-to-thinly-bedded limestone intercalated with calcareous mudstone, and contains a large number of ammonites and bivalves. Conodonts present in bed 14 include: Triassospathodus homeri, Tr. symmetricus, Neospathodus triangularis, Ns. brochus, Ns. curtatus, Gladigondolella tethydis, Cypridodella muelleri, and Neohindeodella triassica. The newly recovered ichthyosauromorph specimen, which we refer to the new genus and species Baisesaurus robustus, accordingly occurs within the Triassospathodus symmetricus-Tr. homeri assemblage zone of the Luolou Formation (Xiang et al., 2020), which is of late Spathian age. Here, we give a detailed description of this specimen, which is an important, morphologically novel addition to the previously known early record of ichthyosauromorphs.

Materials and Methods

The specimen described here was collected by Kunpeng Xiang, Yongzhong He and Housong Zhang from the Guizhou Geological Survey in the course of surveying the regional geology and paleontology of Guangxi. The Guizhou Geological Survey was authorized to carry out a regional geological survey in Guangxi, and to collect fossils as part of their survey work, according to the terms of the entrusted work assignment WT-[2016]-177 from Chengdu Geological Survey Center. The specimen is housed in the School of Earth Sciences, China University of Geosciences (Wuhan), under the collection number CUGW VH107.

CUGW VH107 consists of a single limb element and a disarticulated trunk skeleton including some dorsal vertebrae (12 centra and seven neural arches, most of the neural arches being separated from the corresponding centra), gastralia and ribs. Most of these bones display surface striations where intact, and spongy internal structure where broken (Fig. 2). The disarticulated nature of the skeleton, combined with the good preservation of individual elements, suggests that CUGW VH107 was buried essentially in situ, but that water currents may have disturbed the skeleton after the associated muscle and soft connective tissue had decayed. A large isolated object at the upper left corner of the fossil-bearing slab (Fig. 2) has a curved triangular outline and a rough surface. It is unlikely to be a bone, given the lack of any surface striations or spongy internal structure.

Figure 2 Photograph and outline drawing of Baisesaurus robustus CUGW VH107.

The specimen comprises semi-articulated incomplete vertebrae, a limb element, ribs and gastralia. Abbreviations: Dv, dorsal vertebrae; L, limb element; G, gastralia; R, ribs.

For comparison to CUGW VH107, we calculated ratios of centrum height to centrum length for the dorsal vertebrae of 26 other marine reptile specimens, based on data and images from the literature (see Table S1 in Supplementary Information). Measurements for nine specimens were taken directly from publications, while those for the other 17 taxa were estimated from photos of known scale. The data were graphed as a scatter plot using Excel 2016.

The bone histology of CUGW VH107 was examined using a sample from the distal part of a rib. The sample was embedded in Araldite-2020 one-component resin, cut with an STX-202A diamond wire automatic microtome, and ground to a thickness of about 100 μm with P400, P800, P1000 and P2000 abrasive paper. The polished thin section was then observed under transmitted and polarized light, and photographed using a ZEISS Primotech optical microscope.

Nomenclatural acts

The electronic version of this article in Portable Document Format (PDF) will represent a published work according to the International Commission on Zoological Nomenclature (ICZN), and hence the new names contained in the electronic version are effectively published under that Code from the electronic edition alone. This published work and the nomenclatural acts it contains have been registered in ZooBank, the online registration system for the ICZN. The ZooBank LSIDs (Life Science Identifiers) can be resolved and the associated information viewed through any standard web browser by appending the LSID to the prefix http://zoobank.org/. The LSID for this publication is: urn:lsid:zoobank.org:pub:CE9FCC42-4F06-42A4-B724-AD2AD3E8589A. The online version of this work is archived and available from the following digital repositories: PeerJ, PubMed Central SCIE and CLOCKSS.

Results

Systematic Paleontology

Reptilia Linnaeus, 1758

Diapsida Osborn, 1903

Ichthyosauromorpha Motani et al. 2015

Baisesaurus gen. nov.

Etymology. “Baise” indicates the provenance of the only known specimen of the genus, which was discovered in Zhebao Township in Baise City; -saurus is a common suffix for genus names of fossil reptiles.

Type Species. Baisesaurus robustus sp. nov.

Diagnosis. As for the type and only known species.

Baisesaurus robustus sp. nov.

Etymology. The Latin word robustus means “robust”.

Holotype. CUGW VH107, a partial associated postcranial skeleton.

Locality and horizon. Northwest margin of the Nanpanjiang Basin, Zhebao Township, Longlin County, Baise City, Guangxi, China; upper part of the Luolou Formation, Lower Triassic (upper Spathian).

Diagnosis. A large basal ichthyosauromorph with a unique combination of postcranial characters (* denotes an autapomorphy): anterior dorsal centra deeply amphicoelous, with an average height to length ratio of about 1.2; neural spine long and inclined posteriorly, with the postzygapophyses much more dorsally positioned than the prezygapophyses; parapophyses subrectangular with their anteroventral corners extended to form ventrally directed apices, and situated adjacent to the anterior margins of the centra; diapophyses well developed and semicircular in outline; deep fossae present posterior to the diapophyses*; and radius robust, elongate, and bearing two distal facets, the posterior facet being 80% as wide as the anterior one*.

Description

Vertebrae. Twelve isolated centra and seven recognizable neural arches are preserved (Fig. 3A). The centra are from the anterior dorsal region. The first preserved centrum is exposed in approximate midsagittal section due to breakage, and is strongly amphicoelous (Figs. 3B, 3C). Similarly amphicoelous centra are present in a wide range of other Triassic marine reptiles, including ichthyosauromorphs and sauropterygians (Rieppel, 1998c; Cheng et al., 2012; Takahashi, Nakajima & Sato, 2014; Roaldset, 2017; Ekeheien et al., 2018). The lateral and anterior surfaces of the second centrum are partly missing. However, the lateral surface is exposed in the 3rd, 4th, 6th, 7th, 8th, 9th and 10th centra, the anterior surface in the 11th and 12th centra, and the posterior surface in the 5th centrum. The 10th, 11th and 12th centra are almost completely preserved. The 1st, 4th, 5th, 6th and 7th neural arches are exposed in lateral view, whereas the 2nd and 3rd neural arches are anteroposteriorly compressed and exposed in posterior view. The neural arches are well preserved, but most are separated from their corresponding centra, suggesting a lack of neurocentral fusion (Sander & Faber, 2003).

Figure 3 Photographs of the vertebrae of Baisesaurus robustus CUGW VH107.

(A) All preserved vertebrae, with centra numbered 1–12 in red and neural arches numbered 1–7 in blue, both in order of preservation on the slab; (B) three centra (Nos. 10, 11, 12) and two neural arches (Nos. 6, 7); (C) articulated centra (Nos. 1, 2, 3), showing amphicoelous morphology; (D) isolated neural arch (No. 1) in posterolateral view. Abbreviations: dph, diapophysis; f, fossa; nc, neural canal; nf, neural arch articular facet; poz, postzygapophysis; pph, parapophysis; prz, prezygapophysis.

The lateral surfaces of the centra are subrectangular in outline. The ventral surfaces are smooth, and display a concavity centered on the anteroposterior midpoint (Fig. 3A, centra 6 and 10). The anterior surfaces of the centra have a hexagonal outline, whereas the posterior surfaces are circular (Fig. 3A, centra 5 and 11). Large parapophyses are present on the lateral surfaces of the centra (Fig. 3B). They are subrectangular with their anteroventral corners extended to form ventrally directed apices, and extend anteroposteriorly from the anterior margins of the centra to their mid-regions. The parapophyses of Baisesaurus are similarly shaped to those of Grippia, but unlike those of Chaohusaurus, which have a triangular outline (personal observation, specimen numbers AGM AGB6256 and AGM AGB7401; AGM = Anhui Geological Museum) (Roaldset, 2017; Ekeheien et al., 2018). In hupehsuchians, the parapophyses are always more posteriorly and ventrally positioned (Chen et al., 2014a, 2014b, 2015). In ichthyosaurs such as Cymbospondylus, the centra are shortened, and the parapophyses are much more dorsoventrally elongated than in Baisesaurus (Ekeheien et al., 2018; Engelschiøn et al., 2018). Among some early sauropterygians, the neural arches bear well-developed transverse processes, but parapophyses are weak or absent (Rieppel, 1998a, 1998b; Li & Liu, 2020). The articular facets for the neural arches of Baisesaurus are prominent and rugose, and the floor of the neural canal on each centrum is narrow (Fig. 3C).

Most of the neural arches are separated from the centra, but the arches would generally have articulated with the anterior-middle portions of the centra in life. The neural arches have well-developed prezygapophyses and postzygapophyses extending, respectively, anteriorly and posteriorly (Fig. 3B). The diapophyses of the neural arches are well developed and robust, and extend laterally. The anterior and posterior margins of the diapophyses are straight or slightly concave, whereas the dorsal margins are convex (Fig. 3D). The posterodorsal corners of the diapophyses are prominent and sharply pointed. There are no discernible boundaries between the diapophyses and parapophyses (Fig. 3B), suggesting that these articular facets are combined to form large synapophyses (i.e. apophyses for single-headed ribs). The neural arches contribute significantly to the articular facets as in Chaohusaurus (Huang et al., 2019), but unlike the condition in Omphalosaurus (Sander & Faber, 2003). In hupehsuchians, the diapophyses of the neural arches also contribute to synapophyseal facets, but the facets are positioned more posteriorly than Baisesaurus (Chen et al., 2014b, 2015; Wu et al., 2016). The prezygapophysis is well preserved in the 7th neural arch, and separated from the corresponding diapophysis by a smooth, concave surface (Fig. 3B). A deep fossa is present posterior to the diapophysis (Figs. 3B and 3D). This fossa is not seen in other ichthyosauromorphs or in early sauropterygians, and may represent an autapomorphy. The postzygapophyses are small, paired triangular processes situated on the posterior margins of the neural spines. Their articular surfaces are concave and directed posteroventrally (Fig. 3D). They are much more dorsally positioned than the prezygapophyses, suggesting that the neural spines were inclined posteriorly in order to bring each pair of postzygapophyses into articulation with the prezygapophyses of the following neural arch, as in Chaohusaurus (personal observation, specimen numbers AGM AGB6256 and AGM AGB7401) and Utatsusaurus (Shikama, Kamei & Murata, 1978). In many early sauropterygians the postzygapophyses are also more dorsally positioned than the prezygapophyses, but the neural spines are more elongate and closely spaced, and are not inclined posteriorly (Rieppel, 1998a; Cheng et al., 2012, 2016; Li & Liu, 2020). The neural spines have relatively thin anterior and posterior margins, and are thickest in the central part (Fig. 3D). The anterior margin is slightly concave, whereas the posterior margin has a straight dorsal portion and a concave ventral portion. The facet for articulation with the neural arch extends onto the lateral sides of the centrum, rather than being limited to the dorsal surface (Fig. 3B). The anteroposterior mid-regions of the neural arches protrude ventrally, creating a V-shaped profile in lateral view, and the lateral edges of the dorsal surfaces of the centra have a matching indentation for articulation with the neural arches. The neural arches lack transverse processes.

The preserved anterior dorsal centra have height to length ratios ranging from 1.0 to 1.3, with an average of about 1.2 (Table 2). Such values are comparable to those seen in many basal ichthyosauromorphs, including Utatsusaurus (average ratio of about 1.2, based on 13 anterior dorsal centra) (Shikama, Kamei & Murata, 1978) and Grippia (average ratio of about 1.0–1.1, based on 273 dorsal centra) (Roaldset, 2017), but are slightly larger than the value for Chaohusaurus (about 0.9) (Motani, Mcgowan & You, 1996) and smaller than the values for Omphalosaurus (ratio of 1.6 for a single anterior dorsal centrum) (Sander & Faber, 2003) and all species from the Middle-Late Triassic, Jurassic and Cretaceous (Fig. 4). The ratio of centrum height to centrum length seen in Baisesaurus is also similar to the values for some early sauropterygians, including Chinchenia sungi (Rieppel, 1998b), Sanchiaosaurus dengi (Rieppel, 1998b), Corosaurus alcovensis (Rieppel, 1998a), Augustasaurus hagdorni (Sander, Rieppel & Bucher, 1997) and Yunguisaurus liae (Wang et al., 2019).

Figure 4 Comparison of centra height and length in different marine reptiles.

Black diamond represents Baisesaurus; blue circles represent other Early Triassic non-ichthyosaurian ichthyosauromorphs; green triangles represent ichthyosaurs; red squares represent sauropterygians.

Table 2 Measurements of the preserved centra and neural arches of CUGW VH107.

Centra	
Centrum No.	Max Length (mm)	Max Height (mm)	H/L Ratio	
1	25.7	32.7	1.3	
2	31.7	–	–	
3	25.4	30.1	1.2	
4	29.6	35.9	1.2	
5	30.1	31.1	1.0	
6	26.5	–	–	
7	25.6	29.1	1.1	
8	29.1	32.7	1.1	
9	24.7	30.2	1.2	
10	29.9	31.5	1.1	
11	–	31.7	–	
12	–	30.2	–	
Neural Arches	
Neural Arch No.	Max Length (mm)	Max Height (mm)	
1	14.8	46.9	
2	–	47.9	
3	–	49.1	
4	–	47.3	
5	17.3	47.5	
6	17.5	49.6	
7	18.1	48.8	

Ribs. About 30 recognizable dorsal ribs are preserved in Baisesaurus. Some are almost complete, while others are broken. The ribs are generally long, with lengths ranging from 100 mm to 200 mm, and have relatively thin distal shafts. The ribs are not pachyostotic, even in their proximal portions, and the rib heads are not distinctly expanded and appear trapezoidal in proximal view (Figs. 5A and 5B). The ribs are curved in their proximal and middle parts, but their distal parts are straight. All the ribs are single-headed, and each rib head would have articulated with the synapophysis formed by the centrum and neural arch of the corresponding vertebra, similar to many other early ichthyosauromorphs and to early sauropterygians (Jiang et al., 2014; Cheng et al., 2016; Jiang et al., 2016; Wu et al., 2016; Huang et al., 2019; Li & Liu, 2020). In Utatsusaurus, by contrast, some of the ribs are double-headed (Shikama, Kamei & Murata, 1978). The heads of the ribs are slightly thickened dorsoventrally, but not widened anteroposteriorly. Grooves are present on the anterior and posterior surfaces of the dorsal ribs, giving the rib shafts a figure-eight cross section, but the grooves do not extend distally beyond the middle portions of the rib shafts. Similar grooves are present on the ribs of ichthyopterygians (Sander, 2000). The distal ends of the ribs are small, and round in outline.

Figure 5 Photographs of the dorsal ribs, gastralia and limb element of Baisesaurus robustus CUGW VH107.

(A) Dorsal ribs; (B) rib heads in proximal view (red arrow) and rib shaft in cross-section (black arrow); (C) gastralia; (D) limb element. Abbreviations: l, gastral lateral element; m, gastral median element.

Gastralia. The gastralia of Baisesaurus have been disarticulated, and their original arrangement is unclear. The gastralia include Y-shaped median elements and long, slightly curved lateral elements, and comparisons to other early ichthyosauromorphs suggest that each median gastral element would have been flanked in life by two lateral elements (Figs. 5C and 6F). Each median gastral element consists of two long lateral splints and a long, sharp anterior process, with the lateral splints set at an angle of about 150° to one another and an angle of about 100° to the middle process. The gastralia of Baisesaurus differ from those of sauropterygians, but resemble those of Chaohusaurus (Fig. 6G, H), in having sharp anterior processes. The distance between the distal ends of the paired lateral splints is about 68.5 mm. Each lateral gastral element is a single bar, which is thick in the central part of the shaft and becomes thinner towards the proximal and distal ends (Fig. 5C). About seven complete median gastral elements and more than seven pairs of lateral gastral elements are definitely present. The gastralia are all small, elongate bones with subcircular cross-sections, and are less than a quarter as thick as the ribs.

Figure 6 Drawings of dorsal ribs (A–E) and median gastral elements (F–J) of different marine reptiles.

(A) Baisesaurus robustus (CUGW VH107); (B) ichthyosauriform Utatsusaurus hataii (IGPS95941); (C) sauropterygian Lariosaurus sanxiaensis (HFUT YZS-16-01); (D) sauropterygian Dawazisaurus brevis (NMNS000933-F034397); (E) thalattosauriform Askeptosaurus italicus (PIMUZ T 4832); (F) Baisesaurus robustus (CUGW VH107); (G) ichthyosauriform Chaohusaurus geishanensis (P45-H85-23); (H) ichthyosauriform Chaohusaurus chaoxianensis (AGB6256); (I) sauropterygian Kwangsisaurus orientalis (IVPP V2338); and (J) sauropterygian Hanosaurus hupehensis (IVPP V3231). Yellow represents non-ichthyosaur ichthyosauriforms, and green represents non-ichthyosauromorph marine reptiles. The scale bar which equals 5 cm is for the ribs, and the scale bar which equals 2 cm is for the gastralia.

Radius. A single limb bone is present, and is well preserved with a maximum length of 69.8 mm and maximum width of 32.1 mm (maximum length/width ratio of 2.33). The outline of the limb bone is highly symmetrical in both dimensions, and the proximal and distal ends are both expanded. The exposed surface of the element may be either dorsal or ventral, but in any case is flattened and bears numerous, radially arranged striations (Fig. 5D). One margin of the shaft is straight, whereas the other is concave (Fig. 5D). The proximal end is slightly convex, and the distal end bears two articular facets (Fig. 5D). The smaller one is 80% as wide as the bigger one, and the angle between the facets is about 120°.

The limb element is difficult to identify with certainty given that the rest of the appendicular skeleton is missing, but bears a greater general resemblance to the radii and ulnae of other early ichthyosauromorphs than to their humeri, femora, tibiae or fibulae. In most early ichthyosauromorphs, such as Chaohusaurus, Grippia, Sclerocormus and Cartorhynchus, the humeri and femora are shorter and much more robust than the preserved limb element in Baisesaurus, whereas the tibiae and fibulae are slenderer and more anteroposteriorly asymmetrical (Motani et al., 2015a, 2015b; Jiang et al., 2016; Roaldset, 2017; Huang et al., 2019; Zou et al., 2020). The fact that the other elements preserved in the fossil slab are generally from the anterior part of the trunk also suggests that this element is most likely to be from the forelimb. The limb element is somewhat comparable to the radii and ulnae of hupehsuchians (Figs. 7B–7E), which are long and robust (Carroll & Dong, 1991; Chen et al., 2014b, 2015). However, the radius of Utatsusaurus (Fig. 7I) resembles the limb element of Baisesaurus even more closely, being long, relatively symmetrical in both dimensions, enlarged at both ends, and equipped with a single proximal articular surface and two adjacent distal articular surfaces (Mazin, 1986).

Figure 7 Drawings of forelimb bones of some selected Triassic marine reptiles.

(A) Baisesaurus robustus (CUGW VH107); (B) Parahupehsuchus longus (WGSC 26005); (C) Hupehsuchus nanchangensis (ZMNH M8127); (D) Eohupehsuchus brevicollis (WGSC V26003); (E) Eretmorhipis carrolldongi (WGSC V26020); (F) Chaohusaurus brevifemoralis (AGB7408); (G) Chaohusaurus zhangjiawanensis (WHGMR V26025); (H) Grippia longirostris (PMU R472); (I) Utatsusaurus hataii (IGPS95941); (J) Majiashanosaurus discocoracoidis (AGM-AGB5954) (Sauropterygia); (K) Qianxisaurus chajiangensis (NMNS-KIKO-F044630) (Sauropterygia); (L) Dawazisaurus brevis (NMNS000933-F034397) (Sauropterygia); and (M) Askeptosaurus italicus (MSNM V456) (Thalattosauria). Blue represents hupehsuchians, and other colors represent same categories as in Fig. 6. (B–L) represent left forelimbs, whereas (M) represents a right forelimb. Abbreviations: H, humerus; R, radius; U, ulna.

The limb element differs from the forearm bones of other basal ichthyosauriforms, such as Chaohusaurus and Grippia. In these taxa the radius and ulna are relatively short, expanded at only one end, and/or strongly anteroposteriorly asymmetric (Figs. 7F–7H). The elongate forelimb elements of sauropterygians (Figs. 7J–7L) and thalattosaurs (Fig. 7M) could be considered similar to the Baisesaurus limb element to some degree, but are significantly less robust in the shaft region, and have only a single distal facet. In general, we consider the Baisesaurus limb element to most closely resemble the radius of Utatsusaurus, given that both have a long, robust shaft and a double distal facet, and therefore to be tentatively identifiable as a radius. However, we cannot completely rule out the possibility that the limb element is an ulna, as the radius and ulna are generally similar in morphology in early ichthyosauromorphs.

Bone histology. The distal rib cross-section displays a large medullary cavity (Fig. 8). The cortex is very thin and contains only a few longitudinal vascular canals, as in previously sectioned ribs of Omphalosaurus (Sander & Faber, 2003) and Utatsusaurus (Nakajima, Houssaye & Endo, 2014). Large erosional cavities are apparent in the inner region of the cortex. Parallel-fibered bone is present near the periphery, and a growth line can be seen in the outer cortex (Fig. 8D), suggesting that the only known individual of Baisesaurus was growing slowly near the end of its life (Kolb, Sánchez-Villagra & Scheyer, 2011; Houssaye et al., 2014; Nakajima, Houssaye & Endo, 2014).

Figure 8 Bone microstructure of the distal part of a rib of Baisesaurus robustus CUGW VH107.

(A) Partial cross section showing large medullary cavity and trabeculae; (B) close-up of peripheral area, with white arrows indicating vascular canals; (C) close-up of peripheral area, with white arrow indicating erosional cavity in cortex; (D) close-up of peripheral area, with white arrow indicating line of arrested growth.

Discussion

Baisesaurus is represented only by a partial postcranial skeleton. The centra of Baisesaurus are similar in height-to-length ratio to those of both early ichthyosauromorphs and sauropterygians (Fig. 4). However, this specimen preserves typical ichthyosauromorph characters that preclude referral to other marine reptile groups present in the Early Triassic, including not only Sauropterygia but also Thalattosauria. Vertebral transverse processes are absent in Baisesaurus, as in other ichthyosauromorphs (Motani et al., 2015a), but are well developed in all known Early Triassic sauropterygians (Rieppel, 1998a, 1998b, 1998c; Jiang et al., 2014; Li & Liu, 2020). The dorsal ribs of Baisesaurus are generally thin and are not pachyostotic even proximally, as in ichthyosauromorphs outside Hupehsuchia and Nasorostra (Figs. 6A and 6B) (Young & Dong, 1972b; Shikama, Kamei & Murata, 1978; Sander, 2000; Chen et al., 2013; Takahashi, Nakajima & Sato, 2014; Huang et al., 2019). By contrast, pachyostotic dorsal ribs occur in most sauropterygians (Figs. 6C and 6D), such as Majiashanosaurus (Jiang et al., 2014), Lariosaurus (Li & Liu, 2020), Hanosaurus (Rieppel, 1998c), Serpianosaurus and Neusticosaurus (Rieppel, 1998a). In addition, the gastralia of Baisesaurus include slender median gastral elements with sharp, relatively long anterior processes, a condition seen in other ichthyosauromorphs (Figs. 6F–6H) (Shikama, Kamei & Murata, 1978; Maisch, 2001; Huang et al., 2019). The median gastral elements of sauropterygians are more robust (Rieppel, 1998b; Jiang et al., 2014; Cheng et al., 2016; Li & Liu, 2020), and either bear short, blunt anterior processes (Rieppel, 1998c; Li & Liu, 2020) or lack anterior processes entirely (Figs. 6I and 6J) (Rieppel, 1998b). Thalattosaurs usually have vertebral transverse processes (Nicholls & Brinkman, 1993b; Müller, 2005, 2007), unlike Baisesaurus, and the distal ends of their dorsal ribs are always expanded (Fig. 6E) (Müller, 2005; Rieppel, Liu & Li, 2006; Müller, 2007).

Recent studies have defined the clade Ichthyosauromorpha as comprising the last common ancestor of Hupehsuchus nanchangensis and Ichthyosaurus communis, and all of its descendants (Motani et al., 2015a; Moon, 2019). The diagnostic characters of this clade pertain mainly to the limb elements and neural arches (Motani et al., 2015a). Baisesaurus possess three features that are only seen in Ichthyosauromorpha and may represent synapomorphies of the group: neural arches lacking transverse processes, ribs that are not pachyostotic proximally, and median gastral elements with sharp anterior processes. Among Early Triassic ichthyosauromorphs, Baisesaurus can be excluded from the clades Hupehsuchia and Nasorostra. Although the limb element of Baisesaurus is similar to the forearm bones of hupehsuchians to some degree, the axial skeleton displays clear differences from the hupehsuchian condition. The neural arches of hupehsuchians are tall, with bipartite neural spines that each comprise two vertically stacked segments, and their dorsal ribs are always expanded, having anterior and/or posterior flanges (Carroll & Dong, 1991; Chen et al., 2014b, 2015). Similarly, Baisesaurus differs from Nasorostra (i.e., Sclerocormus and Cartorhynchus), in which the trunk is heavily built with thickened ribs, symmetrical median gastral elements are absent, and the zeugopodial bones are short and robust (Motani et al., 2015a; Jiang et al., 2016). Baisesaurus may be a member of the clade Ichthyopterygia (Jiang et al., 2016; Moon, 2019), but clear ichthyopterygian or even ichthyosauriform synapomorphies are absent in the available partial skeleton. Accordingly, we regard Baisesaurus at present as a basal ichthyosauromorph of uncertain phylogenetic position, although definitely not a member of Hupehsuchia or Nasorostra.

Early Triassic ichthyosauriforms can be divided into two morphological categories, as mentioned in the introduction (Zou et al., 2020). Assuming the bone we tentatively identify as the radius of Baisesaurus really is from the forearm, then Baisesaurus is comparable to Zou et al. (2020) Type A ichthyosauriforms in having elongate zeugopodial elements, and differs from Chaohusaurus chaoxianensis (and perhaps also C. brevifemoralis) in having more robust forearm bones. Baisesaurus is also distinguished from most basal ichthyosauriforms (including Chaohusaurus, Parvinatator, Grippia and Gulosaurus) by larger body size and some morphological details of the centra. For example, the parapophyses of Baisesaurus are subrectangular with ventrally directed apices, whereas those of Chaohusaurus are triangular (personal observation, specimen numbers AGM AGB6256 and AGM AGB7401). The parapophyses are less well developed in Grippia (Roaldset, 2017; Ekeheien et al., 2018) than in Baisesaurus. Ichthyosaurs are represented in the Lower Triassic by some large Cymbospondylus and Pessopteryx specimens from Spitsbergen (Engelschiøn et al., 2018), but like other members of the highly derived clade Ichthyosauria (Moon, 2019), these taxa differ from Baisesaurus in many respects. For example, they possess significantly shortened centra, and shortened forearm bones.

Baisesaurus shares more similarities with Utatsusaurus hataii from Japan than with any other previously reported ichthyosauromorph. Both Baisesaurus and Utasusaurus are relatively large, some specimens of the latter being close to 300 cm in length (Shikama, Kamei & Murata, 1978; Sander, 2000). The limb element of Baisesaurus is also very similar to the radius of Utatsusaurus (Fig. 7), as both are long, robust, highly symmetrical, and endowed with two distal articular facets (Shikama, Kamei & Murata, 1978; Mazin, 1986). There are also some differences between the two taxa. The ribs of Utatsusaurus are gently broadened distally, and some of them are double-headed (Shikama, Kamei & Murata, 1978) whereas the preserved ribs of Baisesaurus are all single-headed and have unexpanded distal ends. In Utatsusaurus the proximal end of the radius is flat and the more posterior of the two distal facets is much smaller than the anterior one (Mazin, 1986), whereas in Baisesaurus the proximal end of the possible radius is convex and the two distal facets are similar in size, the posterior facet being 80% as wide as the anterior one.

The specimen CUGW VH107 comprises only some remains from the anterior part of the trunk, but represents the first Early Triassic ichthyosauromorph to be discovered in Guangxi. The specimen differs in important respects from all previously known Early Triassic ichthyosauromorphs, displaying the following autapomorphies: deep fossae present posterior to the diapophyses; and robust radius bearing two distal facets, the posterior facet being 80% as wide as the anterior one. CUGW VH107 is also the largest Early Triassic ichthyosauromorph so far discovered in China, with an estimated body length about two times greater than that of Sclerocormus, which was previously considered to be the largest ichthyosauromorph known from the Early Triassic of China (Jiang et al., 2016). For these reasons, and despite the incompleteness of the specimen, we consider CUGW VH107 to clearly represent a new, large basal ichthyosauromorph, Baisesaurus robustus, from the Lower Triassic of South China. Based on the single preserved limb element, Baisesaurus can be tentatively inferred to have had elongate, compactly arranged forelimb bones, and to have been a strong swimmer capable of dispersing over long distances (Zou et al., 2020). In this respect, Baisesaurus was likely similar to Utatsusaurus, which occurs in both Japan (Shikama, Kamei & Murata, 1978) and Canada (Nicholls & Brinkman, 1993a).

Baisesaurus is much larger than Chaohusaurus, and comparable in body size to Utatsusaurus. The 10 dorsal centra of Baisesaurus whose lengths can be measured have an average length of 27.8 mm, whereas in Chaohusaurus the length of any single centrum is less than 10 mm (Young & Dong, 1972c; Huang et al., 2019). Nevertheless, Chaohusaurus and Utatsusaurus are two well-studied basal ichthyosauromorph taxa, so their body proportions can provide a reliable basis for estimating the total length of an intact Baisesaurus. In the holotype (AGB7401) of C. brevifemoralis (Huang et al., 2019), the average length of the anterior dorsal vertebrae is about 7 mm, and the total body length is about 80 cm. Assuming similar skeletal proportions, the total length of Baisesaurus would be about 318 cm. Similarly, the length of a single anterior dorsal vertebra in the holotype (specimen No. K1, collection number IGPS 95941) of U. hataii (Shikama, Kamei & Murata, 1978) is about 13.8 mm, and the total body length is estimated as 140 cm. Based on the proportions of U. hataii, the estimated length of Baisesaurus would be about 300 cm. With a total body length of about 3 m in life, Baisesaurus was relatively large among basal ichthyosauromorphs (e.g., Chaohusaurus, 60–100 cm; Sclerocormus, 160 cm; Utatsusaurus, 140–300 cm; Gulosaurus and Grippia, estimated at less than 120 cm) (Wiman, 1910; Young & Dong, 1972c; Shikama, Kamei & Murata, 1978; Sander, 2000; Cuthbertson, Russell & Anderson, 2013; Jiang et al., 2016; Huang et al., 2019).

Conclusions

Baisesaurus represents a newly reported large basal ichthyosauromorph from the Lower Triassic of South China. Among previously described taxa, Baisesaurus is most similar to Utatsusaurus, and differs from other, smaller Chinese early ichthyosauromorphs. Baisesaurus is the first Early Triassic ichthyosauromorph to be reported from Guangxi. Together with Cartorhynchus (Motani et al., 2015a), Sclerocormus (Jiang et al., 2016) and two recently discovered species of Chaohusaurus (Chen et al., 2013; Huang et al., 2019), Baisesaurus expands the known geographical distribution of early ichthyosauromorphs in China. Alongside other recent finds, Baisesaurus implies a need for further studies of the paleoecology and paleogeography of Early Triassic ichthyosauromorphs in order to understand their role in the ecosystem at the eastern margin of the Paleo-Tethys. The discovery of Baisesaurus has considerable paleogeographic significance, as well as adding to the documented taxonomic diversity of South China’s ichthyosaur fauna in the Spathian interval of the Early Triassic.

Supplemental Information

Supplemental Information 1 The extra references used for comparison in Figures 4, 6, 7.

Click here for additional data file.

We thank the Guizhou Geological Survey and Chengdu Institute of Geology and Mineral Resources for collecting CUGW VH107, Dongyi Niu and Yan Chen for preparing this fossil, and Rui Wu for helping to prepare the histological sections described in this paper. We thank Ryosuke Motani for very useful discussion and comments. Martin Sander and Dayong Jiang also provided very useful suggestions.

Additional Information and Declarations

Competing Interests

Author Contributions

Data Availability

New Species Registration

The authors declare that they have no competing interests.

Jicheng Ren conceived and designed the experiments, performed the experiments, analyzed the data, prepared figures and/or tables, authored or reviewed drafts of the paper, and approved the final draft.

Haishui Jiang conceived and designed the experiments, authored or reviewed drafts of the paper, and approved the final draft.

Kunpeng Xiang conceived and designed the experiments, authored or reviewed drafts of the paper, and approved the final draft.

Corwin Sullivan analyzed the data, authored or reviewed drafts of the paper, and approved the final draft.

Yongzhong He conceived and designed the experiments, authored or reviewed drafts of the paper, and approved the final draft.

Long Cheng conceived and designed the experiments, analyzed the data, authored or reviewed drafts of the paper, and approved the final draft.

Fenglu Han conceived and designed the experiments, performed the experiments, analyzed the data, prepared figures and/or tables, authored or reviewed drafts of the paper, and approved the final draft.

The following information was supplied regarding data availability:

The raw measurements are available in the main article, the extra references used for comparison in Figures 4, 6, 7 are available in the Supplemental File.

The following information was supplied regarding the registration of a newly described species:

Publication LSID: urn:lsid:zoobank.org:pub:CE9FCC42-4F06-42A4-B724-AD2AD3E8589A

Genus, Baisesaurus Ren LSID: urn:lsid:zoobank.org:act:50BFE208-C9DA-4769-A1F3-2012BE514689

Species, Baisesaurus robustus LSID: urn:lsid:zoobank.org:act:ADF531F2-F910-4B84-95B4-FAC5E15DB8E5.

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
