# Peer review of "A new basal ichthyosauromorph from the Lower Triassic (Olenekian) of Zhebao, Guangxi Autonomous Region, South China"

_PeerJ, doi:10.7717/peerj.13209_

## Round 0.1 · original submission · Major Revisions

Dear authors,

Thank you for your submission to PeerJ and congratulations on your discovery. Following advice from two expert reviewers, I recommend major revisions. With your revised submission, please submit a clean version of your manuscript, a tracked changes version showing all changes made, and an itemized response to reviewers where you respond to each reviewer recommendation.

There are a number of main points that should be addressed in your revision.

1) Both reviewers noted substantial English grammar issues, which make the meaning of parts of the manuscript difficult to interpret. Unfortunately PeerJ does not offer English editing services as standard, but I would strongly recommend you have an English colleague read through the manuscript and correct it for English after you have revised the manuscript for content.

2) Both reviewers had doubts about the anatomical accuracy of some of your assertions, particularly that of the femur. Please see the reviewer comments for details.

3) In addition, reviewer 1 is concerned about the identification of the specimen as an ichthyosaur. As you go through their review in detail, please add more support to your assertion that your specimen is an ichthyosaur.

Thank you again for your submission. Please let me know if you have any questions.

Best,

Brandon P. Hedrick, Ph.D.

·

Basic reporting

The first and foremost suggestion that I have is that the article is thoroughly checked by a native (preferrably two) English speaker with a geological/palaeontological background. I am not one myself, but I have encountered multiple, largely minor, but also many major issues with the use of the English language throughout the entire article. I have marked them only in the abstract, because there are so many mistakes that it is futile to try and correct everything at the current state of the MS. I would have to re-write it basically. The article would greatly benefit from such an improvement, which can be easily achieved. I consider this mandatory before re-submission.

Concerning literature references I would strongly suggest to include pertinent information about the biostratigraphy of the Luolou Formation, which is available concerning both ammonite and conodont zonation. The authors provide an absolute date published recently by Xiang et al. (2020). That is fine, but in order to compare the specimen to other lower Triassic marine reptiles it is not very helpful. Biostratigraphic data would be much more relevant, and the available recent literature should be cited. I Would also like to know more about the general fanual composition of the formation, at least some cornerstone data, to get an idea if and to what extent the paleoecosystem differs from, e.g. that of the coeval Nanlinghu Formation of Anhui, which so far has yielded the huge majority of early Triassic ichthyosauriform finds from China.

The structure of the article is basically fine, but it is at times close to unreadable due to the very poor English.

The article is, of course, self-contained, as it describes a single new fossil specimen of a (presumed) ichthyosaur from a locality where such fossils were hitherto unknown (but, of course, quite expectable considering the facies and age of the formation). Its main hypothesis is that the new find indiactes the presence of a hitherto unknown, moderately large growing Lower Triassic ichthyosaur. I consider the arguments provided by the authors for an ichthyosaurian identification insufficient, as pointed out more clearly below. They have to do a lot of work to improve that.

Experimental design

The article certainly represents a piece of original primary reserach. The fossil specimen described has never been published before and it fills a gap in the fossil record of marine reptiles from China. It also indicates the possible presence of a hitherto unrecognized taxon, but this is far from certain.

The research question is basically the description, taxonomic allocation and comparison of the new fossil find to already known taxa of early Triassic ichthyosauriforms. I consider that the authors have failed in three major points:

1. The identification of the specimen as an "ichthyosaur" rests solely on symplesiomophies (amphicoelous vertebrae, synapophyseal rib articulation etc.) which are shared by a number of other Triassic marine reptile taxa, such as more basal Ichthyopterygians and ichtyosaurifroms, hupehsuchids, thalattosaurs etc. The authors do not identify a single apomorphic character which links their fossil exclusively to ichthyosaurs, or at least any particular ichthyosaurian taxon.
I particulary do not understand why they insist on it being an ichthyosaur. There are different usages of the "Ichthyosauria" in the literature. I, e.g., consider froms such as Utatsusaurus and Chaohusaurus as "ichthyosaurs" and see no reason for further subdivsion. That is a minority report, however. Most authors follow the nomenclature proposed by Motani and co-workers, where the Ichthyosauria is basically restricted to cymbospondylids, mixosaurids and shastasaur-grade ichthyosaurs and the euichthyosaurs among Triassic forms, whereas more basal taxa, down to the hupehsuchians, are classified as ichthyoptergians, ichthyosauriforms and ichthyosauromorphs.
It appears highly unlikely that the animal described is a cymbospondylid, mixosaurid, shastasaur or euichthyosaur. It should teherfore be best identified as an "ichthyosauromorph" until a more detailed, apomorphy-based identification can be provided.
To amend this I strongly suggest that the authors do a more detailed comparison of their find not only to ichthyopterygians, but also to other Triassic marine reptiles and try to find synapomorphic features which provide a better basis for their identification. It will then also be possible to find support for a placement of the specimen in Ichthyosauromorpha, Ichthyosauriformes or maybe Ichthyopterygia (quite certainly, however, not Ichthyosauria, as it does not show a single ichthyosaurian character (in the sense of Motani and co-workers) as far as I can judge it from the description and illustrations)

2. The anatomical identifications are not without doubt. I have raised suspicions that the femur, upon which a lot of the discussion and comparison provided by the authors rests, is probably another element. This is indicated by morphology, particularly the apparent lack of a well-defined caput femoris. Instead the proximal portion of the femur is a completely flattened plate, even though the specimen, as seen from the three-dimensional vertebrae an neural arches, is almost uncompressed otherwise. It also is slightly indicated by taphonomy, It would be surprising to find a hind limb element when the rest of the skeleton is clearly from the anterior region of the body.
I consider the identification as a radius at least as viable as the identification as a femur Again the authors have to do more comparative work to other early ichthyosauriforms and Triassic marine reptiles, including not only the femur but also the zeugopodial elements of the fore- and hind fins. The bone is certainly not a humerus, and I also cannot place it in the shoulder or pelvic girdle of any marine reptile, but the hypothesis that it is a zeugopodial element instead of a femur is not excluded by any of the arguments they provide. Double articulations at the distal end are not exclusive to the femur.
I also have the strong suspicion that something went wrong in the description of the neural arches. What the authors identify as "paired posterior processes" halfway up the posterior margin of the neural arch are, in my opinion, simply the paired postzygapophyses. What they label as postzygapophysis must be some other structure. It could be a well-developed hyposphene. If the animal had a hyposphene-hypantrum articulation, this would greatly affect both its anatomy and the potential taxonomic dentifications. It is mandatory that this question is addressed and sufficiently clarified by the authors.

3. There is no detailed comparison to Sclerocormus parviceps Jiang et al. 2016, which is from the Lower Triassic (Nanlinghu Formation, Spathian, Olenekian) of Anhui. The type skeleton is about 1.5 meters long, so it is the only Lower Triassic ichthosauiform form China currently named and described that at least comes close in size to the new find. Also it is geographically and stratigraphically very close. Also comparison is too limited to the equally roughly coeval and geographically close hupehsuchids, which show considerable similarities to ichthyosaurs in many features of their postcranial skeleton and have been classified by the team around Prof. Motani and Prof. Jiang as ichthyosauromorphs, probably correctly so.

I consider the investigations carried out by the authors as generally of good quality. they try to give a very deteiled anatomical description, provide detailed mesurements of the skeletal elements and even performed a paleohistological investigation on the specimen. They have certainly tried their best to squeeze out as much relevant information as such an imperfect and disarticulated specimen can offer. I see no ethical problems whatsoever, They have completely adhered to all scientific ethical standards.

The methods of the authors are basically standard methods of descriptive systematic palaeontology, and these are competently used.

Validity of the findings

The paper offers the description of a novel fossil from an as yet "white spot" on the map, at least regarding ichtyhosaurs and their close relatives. Findings of these animals from the early Triassic are rare worldwide and restricted to a limited number of localities, so any substantial addition to their fossil record is welcome. The specimen is well preserved, although incomplete, and described in considerable detail with good figures. I reckon that the study will be cited by experts in the field (students of Triassic marine reptiles, particularly ichthyosaurs and related forms) and probably also by students of local geology and paleontology.

The underlying data have been provided to the author's best ablities. The specimen is documented by good photographic illustrations and informative drawings, they have provided detailed measurements of the relevant skeletal elements, documented the paleohistological sections with good color photograhs and also have documented the geographic and stratigraphic provenance of the find adeuqately (although, as pointed out above, biostratigraphic data need to be addressed in more detail and with additional references).

The authors provide logical and coherent conclusions. My concerns are rather with the basic assumptions (i.e. that the fossil represents an "ichthyosaur", which in the stricts sense of Motani and co-workers it certainly doesn't. Even that it is an ichthyosauromorph is, in my opinion, not adequately supported by the available data, or at least the authors failed to provide sufficent evidence to support their premises.

The article does not indugle into speculation of any kind. That is always welcome, particularly when dealing with incomplete fossil specimens as the one under consideration.

Additional comments

Daer authors

I congratulate you on your findings. But please consider the following:

1. Please let your MS be checked by one or preferrably two native English speakers or colleagues with outstanding knowledge of the English language. They should also be familiar with geology and paleontology, because I have made the experience that native speakers not familiar with the topics tend to introduce more mistakes into a text than they help to eliminate. In its present state the MS is unacceptable to any international journal particularly because it fails to meet the language standards. This can easily be amended by the kind help of one or two colleagues.

2. Please provide a better justification for the identification of the material. I do not consdider it to be an ichthyosaur, at least not in the sense Ichthyosauria has been used in recent years by the majority of students. It may be an ichthyosaur in the sense of my own studies, but these are not the current cladistic consensus. You even do not provide sufficent evidence that the specimen is an ichthyosauromorph (sensu Jiang, Motani and co-workers), i-e. belongs to the clade of hupehsuchians and ichthyosauriforms. The features that you cite to support your identification are symplesiomorphies, even shared with totally different marine reptiles such as thalattosaurs (amphicoelous vertebrae, synapophyseal rib articulation etc.). Please extend your comparisons and try to identify apomorphic features that link your specimen to ichthyosauromorpha or one of its subclades.

3. Re-consider your identification of the femur. it may be one, but i have problems, particularly as I cannot see any indication of a caput femoris (the articular head of the femur which should form a functonal joint with the acetabulum). Your sepcimen appears to be little diagenetically compressed, the vertebrae and neural arches being almost 3D in preservation. The "femur" however is a completely flattened, plate-like bone. i would suggest such a morphology to be found rather in a zeugopodial limb element (radius/ulna or tibia/fibula) than in a stylopodial one (humerus/femur). You should extend comaprison to ther limb elements of basal ichthyosauromorphs and maybe additional marine reptiles, because I also think you may not be right concerning the morphology of the neural arches.
I suggest that your "posterior paired process" of the neural arch is the postzygapophyses, still paired, as usual in basal ichthyosauromorphs. What you label as postzygapophysis appears to be a triangular posterior process that is very reminiscent of the accessory intervertebral articulations found in some non-ichthyosauromorph marine (and many other) reptiles. It suggests the presence of a functional hyposphene-hypantrum articulation in your animal. The potential taxonomic implications of this are far-reaching, and you need to address this question in detail.

4. Please provide additional comparative data on Sclerocormus parviceps in particular, which is the only other large-growing ichthyosauriform currently described from the same general area and stratigraphic level. Also compare more closely to hupehsuchians and, if the hyposphene-hypantrum hypothesis I formulated above should play out, also to non-ichthyosauromorph marine reptiles.

5. Please provide some additonal data on the fossil content, possible paleoecology and particularly the biostratigraphy of the Luolou Formation. There are studies on ammonites and conodonts, and it should be possible to give precise biostratigraphic data for the specimen. Please provide these, as well as the important references. This will geratly facilitate biochronological comparison to other early Triassic marine reptile finds and localities worldwide.

With best regards

Michael W. Maisch

Reviewer 2 ·

Basic reporting

This is a very interesting and important find.
However, I am not sure if the identification ofthe long bone lement as femur is correct and the histological study performed is not suitable to give an appropriate age estimation (also there is morphological evidence for an early ontogenetic stage/not fully grown individual).
The engl. neds some improvement some formulations read strange and authors mixed Singular and Plural...
Sincerely
NK

Experimental design

no comment

Validity of the findings

no comment/see above basic reporting

Additional comments

see above basic reporting and attached pdf

Annotated reviews are not available for download in order to protect the identity of reviewers who chose to remain anonymous.

---

## Round 0.2 · Major Revisions

Dear authors,

Thank you for your resubmission. Unfortunately I was not able to get one of the original reviewers to re-review your paper. However, I was concerned about the taxonomic ID issues that that reviewer brought up so I decided to ask an additional reviewer to take a look. They had the same issues and are not convinced that the specimen is an ichthyosauromorph. I think that this will require some substantial revisions to your paper, either by adding more evidence that the specimen is an ichthyosaur or by altering your taxonomic conclusions.

Please let me know if you have any questions and I would be happy to answer them. Thank you for your submission to PeerJ.

Best,

Brandon P. Hedrick, Ph.D.

Reviewer 2 ·

Basic reporting

The authors addressed most of the issues I had pointed out accordingly, except for the histological part.
The authors state that they had discard the part of age estimation but already in the abstract they state:
“The specimen was not fully grown at the time of death, based on the bone histology of a dorsal rib.”

In line 156 they wrote:
“to assess the growth stage the ichthyosauromorph had reached at the time of death.”

Based on the sample they have (distal part of a likley largely cartilaginous rib) they cannot say anything about age (see Klein et al. 2019; ref. given in the 1st review) and they should delete/rephrase this.
As pointed out in the 1st round, I agree that the specimen was not fully grown but by refering to morphology and not histology.

In Fig. legend 5 they should delet inner (inner cortex), this is far from inner cortex.
Except for this I have no other concerns.

Experimental design

no comment

Validity of the findings

no comment

Additional comments

no comment

·

Basic reporting

Dear editor and authors,
Thank you for the opportunity to review this paper. Early Triassic marine reptiles are both interesting and important. The fossil presented here is no doubt an interesting fossil with a good preservation from an important time slice.

My major concern and the reason for recommending a rejection of the manuscript in its present state, is that I do not believe that the taxonomic conclusion is correct: this is likely not an ichthyosauriform or ichthyosaur. The fossil is presented as an ichthyosauromorph (the more inclusive group, also including hupehsuchians), and I cannot rule out the possibility that this is a hupehsuchian. However, the entire framing of the paper, the discussion and context (as well as the species list) places this fossil closest to the ichthyosaurs, which I strongly doubt. I agree there are some similarities with Sclerocormus parviceps, but also with sauropterygians and hupehsuchians. It might also be necessary to consider additional reptile groups.

Because I think the entire framing of the fossil is based on an incorrect taxonomic placement, I will not comment upon details in the manuscript, as it will have to be rewritten completely after it has been made clear what the specimen actually is. With regard to the text on ichthyosaurs, there are several mistakes and sentences that need clarification and references to the newest literature.
My main reasons for not believing this is an ichthyosaur or close relative:
- The vertebrae are relatively long compared to height, compared to ichthyosaurs, where they are a lot anteroposteriorly shorter, including the earliest taxa from China. I suggest calculating the relationships between height, length and width for comparison to different groups.
- The limb bone differs from other, known ichthyosaur limb bones. I acknowledge the discussion about this element, and it might be an ulna or radius, but it likely belong to a different animal group altogether.
- The ribs and neural arches are also not very typical of ichthyosaurs and close relatives.

Experimental design

In order to present the fossil as an ichthyosauromorph, the authors would need to build a much stronger case, based on cladistics characters and recent phylogenies. The characters given in the manuscript for inclusion within Ichthyosauromorpha are not those given in Motani et al 2015 defining the clade, nor do they refer to any other paper revising the entire group. The most recent full phylogenetic analysis of ichthyosaurs, Moon 2017, is not referred, nor is a paper citing the amphicoelous vertebrae as a defining feature (this could have been Sander 2000). Single similarities to specific ichthyosaur/ possible ichthyosaur taxa eg Omphaloasurus and Grippia do not build an argument for inclusion in ichthyosauromorphs. Huang et al 2019 is a very interesting paper, but they do not provide a character-based definition of Ichthyosauromorpha, and should not be used as such.

I appreciate the combination of outer skeletal anatomy and inner bone microstructure, but there is not sufficient knowledge about this to be used as characters for placing the specimen in a given clade. Similarities to Omphalosaurus or Mixosaurus in the microstructure mentioned in the paper are features shared widely in many vertebrates, and should at the present not be used to draw taxonomic conclusions.

I also think the pictures of the histological sections of the fossils should be improved, and a discussion is lacking on whether the large, internal cavity might actually result from decomposition rather than being an actual feature.

Validity of the findings

See comments above.

Additional comments

As I think the fossil needs to be reanalyzed again for correct taxonomical placement, I do not comment on all of the text in this manuscript. However, I want to make a note on the material from Spitsbergen, which is something I have worked on. The reference Roaldset 2017 is a Master thesis, meaning it is not peer-reviewed. I strongly recommend instead to refer to Engelschiøn et al 2018: Large-sized ichthyosaurs from the Lower Saurian niveau of the Vikinghøgda Formation (Early Triassic), Marmierfjellet, Spitsbergen in Norwegian Journal of Geology, as it is a far more comprehensive and correct discussion of the taxa from Spitsbergen.

---

## Round 0.3 · Minor Revisions

Dear authors,

Thank you for your careful responses to the previous round of reviews. The manuscript is much improved. One reviewer had a few small changes that would improve the manuscript further. After addressing their suggestions, I think the manuscript will be publishable in PeerJ.

Specifically, I think it would be useful to add some more justification as to why you think it is a good idea to name the specimen given that it is largely vertebral remains. Noting that the ratios also line up well with sauropterygians in the text is also important.

Additionally, I found a few small changes upon reading this draft:

Line 85: subage?

Line 162: Although you say you identify it as Baisesaurus in the abstract, you don’t say it in the introduction. Maybe change this sentence to: ‘The specimen that we have identified as a new species, Baisesaurus, accordingly occurs…’

Line 403: ‘stronger forelimb bones’

Line 462: delete comma between ‘…ichythosauromorphs in order to…’

Thank you for your submission to PeerJ. I am happy to answer any questions you may have.

Best,

Brandon P. Hedrick, Ph.D.

·

Basic reporting

Basic reporting

Dear editors and authors,
Thank you for the opportunity to review this manuscript for a second time. The article has gone through a substantial change since the previous round of review, and significantly improved.

My major concern in the previous review was the taxonomic placement of the new specimen. I am happy to see that this has been taken seriously in the new version of the manuscript. I am still not 100% convinced that this is an ichthyosauromorph, but I find that the authors have explored the question thoroughly. I very much appreciate the work on the vertebral ratios, however it should be pointed out that the values for the new specimen might just as well belong to a sauropterygian, judging from the plot, so this should be mentioned.

Specific comments:
Introduction:
- Please have a second look at the paragraph about the Spitsbergen material. There are some inconsistencies, especially with regard to Pessopteryx

Geological setting
- I suggest making a separate heading for Geological setting, as is often done for this type of papers (but this is really up to the editor/ journal style)
- Mention clearly that this is a marine setting
- I would also recommend adding an age in Mya for the specimen, in addition to the late Spathian age, if that is available

Material and methods
- A section on the rationale and method behind the calculation on vertebral ratios is missing. This should also include (possibly in the supplementary) a list on the specimens used in this analysis

Description
- I miss comparisons to other taxa in the first two paragraphs of the section, on the morphology of the vertebrae
- In the paragraph on neural arches, please state clearly that they are separate from the centra
- In all “personal observation”, please add a specimen number
- Numbers for vertebral ratios are given with two decimals, which I think are too many with regard to how detailed measurements are possible in such cases
- As mentioned above, the ratio for the centra of the new specimens are not only close to ichthyosauromorphs in the plot, but also sauropterygians. This should be mentioned. You could also consider doing a statistical analysis to put a number on this. In any case, I suggest removing the trend lines for each group.
- Gastralia: I appreciate the figure for gastralia in other taxa. It should be referenced in the text on gastralia
- Radius; the sentence beginning with “Most early ichthyosauromorphs”, please add taxon names and reference

Discussion
- Please add a discussion on the most recent diagnosis for the clade Ichthyosauromorpha, also if there might be need for a revision of this
- I also suggest stating why it is chosen to erect a new genus and species based on what is mainly vertebral remains (no skull and only one limb element) – this is not always adviced, so I suggest explaining more clearly why you see this as the right thing to do here

Experimental design

see pt 1

Validity of the findings

see pt 1

Additional comments

see pt 1

---

## Round 0.4 · accepted · Accept

Dear authors,

Thank you for your careful attention to comments from the previous round of review. I now find that this manuscript is publishable in PeerJ. There were just two additions that should be made prior to publication, but I want to move this paper to the next stage.

Line 196: Add the Excel version number and year

Line 445: Don’t capitalize ‘introduction’

Thank you for your submission to PeerJ. Don't hesitate to reach out if you have any questions.

Brandon P. Hedrick, Ph.D.